# Genetic analysis of QTLs controlling allelopathic characteristics in sorghum

Tariq Shehzad[1¤]*, Kazutoshi Okuno[2]

1 Plant Genome Mapping Laboratory, College of Agricultural and Environmental Sciences, University of Georgia, Athens, Georgia, 2 Kayada 2069-6, Yachiyo, Chiba, Japan

¤ Current address: Department of Biological and Environmental Sciences, College of Arts and Sciences, Qatar University, Doha, Qatar
* tshehzad@qu.edu.qa

**Data Availability Statement:** All relevant data are within the manuscript and its Supporting Information files.

**Funding:** This research was financially supported under Japan Society for the Promotion of Science

## Abstract

Mature sorghum herbage is known to contain several water-soluble secondary metabolites (allelochemicals). In this study, we investigated quantitative trait loci (QTLs) associated with allelochemical characteristics in sorghum using linkage mapping and linkage disequilibrium (LD)-based association mapping. A sorghum diversity research set (SDRS) of 107 accessions was used in LD mapping whereas, $F_{2:3}$ lines derived from a cross between Japanese and African landraces were used in linkage mapping. The QTLs were further confirmed by positional (targeted) association mapping with Q+K model. The inhibitory effect of water-soluble extracts (WSE) was tested on germination and root length of lettuce seedlings in four concentrations (25%, 50%, 75% and 100%). A Significant range of variations was observed among genotypes in both types of mapping populations ($P < 0.05$). A total of 181 simple sequence repeats (SSRs) derived from antecedently reported map have been used for genotyping of SDRS. A genetic linkage map of 151 sorghum SSR markers was also developed on 134 $F_2$ individuals. The total map length was 1359.3 cM, with an average distance of 8.2 cM between adjacent markers. LD mapping identified three QTLs for inhibition effect on germination and seven QTLs for root length of lettuce seedlings. Whereas, a total of six QTLs for inhibition of germination and ten QTLs for root length were detected in linkage mapping approach. The percent phenotypic variation explained by individual QTL ranged from 6.9% to 27.3% in SDRS and 9.9% to 35.6% in $F_{2:3}$ lines. Regional association analysis identified four QTLs, three of them are common in other methods too. No QTL was identified in the region where major gene for sorgoleone (SOR1) has been cloned previously on chromosome 5.

## Introduction

Every living organism interacts with others and form an ecological system on the earth. Such interactions between organisms at several levels have been long known and studied. Allelopathy is a phenomenon observed in large number of plants that emit specific chemicals interacting on other organisms, including animals and microorganisms, in either inhibitory or

((https://www.jsps.go.jp/english/e-ippan/index. html) postdoctoral fellowship for overseas researchers FY2010-2011 to TS. We are also thankful to Qatar National Library (QNL) for paying all article processing charges to journal. The funders had no role in study design, data collection and analysis, decision to publish, or preparation of the manuscript.

**Competing interests:** The authors have declared that no competing interests exist.

excitatory way. The use of traditional methods to control weeds are often very costly, labor intensive and dependent on weather conditions. Likewise, the extensive utilization of chemicals for controlling weeds may cause serious environmental hazards. Recently, in wheat (*Triticum aestivum* L.), sorghum allelopathic characteristics has been exploited as a substitute for chemical herbicides to reduce environmental pollution. Mature sorghum herbage contains a number of water-soluble chemical compounds (allelochemicals) that has the tendency in controlling weeds. Even the residues of sorghum minimize the incidence and growth of a number of weed species such as purslane and smooth crabgrass [1], green foxtail, velvetleaf and smooth pigweed [2, 3]. Incorporation of herbage of mature sorghum plants into the soil at 2 to 6 ton/ ha and a spray of water extracts reduces the occurance of weeds and increase the yield of irrigated wheat as reported in Cheema et al. [4]. The interplanting of sorghum is observed to reduce the growth and biomass of important weeds of different crops [2]. Sorghum residues release many allelochemicals such as sorgoleone, cyanogenic glycosides-dhurrin and many other products due to the breakdown of large chemical compounds. All these chemicals plays an important role in weed suppression and thus, can be affectively utilized as natural control of major weeds in crops. Sorghum has a great potential for the use as an allelopathic crop to manage parasitic weeds, particularly *Striga* and *Orobanche* species that pose a significant risk to agriculture because of difficulty to deal with them.

Sorghum is producing phytotoxins such as the potent benzoquinone sorgoleone (2-hydroxy-5-methoxy-3- [(8$^{/}$Z, 11$^{/}$Z)-8$^{/}$, 11$^{/}$, 14$^{/}$-pentadecatriene]-*p*-benzoquinone) and its analogs, generally depicting the principal constituent of *Sorghum bicolor* root exudates. Until most recently, a single gene *SOR1* has been cloned which is associated with the production of sorgoleone in sorghum [5].

Sorgoleone is one of the most investigated allelochemicals [5–10]. It was first discovered by researchers studying secondary metabolites affecting germination of parasitic weed *Striga asicatica* (witchweed) [11]. The biosynthesis of sorgoleone has been elucidated using retrobiosynthetic NMR analysis [12, 13]. Sorgoleone is absorbed through the hypocotyl and cotyledonary tissues of growing seedlings, consequently impeding the process of photosynthesis. The mode of action of sorgoleone is similar to soil-applied herbicides because this phytotoxic exudate is also discharged into the soil instantaneously through the root nodules. Most importantly, the effect of sorgoleone into the soil may sustain for longer period than that of applied herbicide.

Recently, DNA marker-assisted selection has been implied to identify QTLs responsible for the production of allelochemicals in various crops including rice and rapeseed. In sorghum, a lot of work has been done to explore the biosynthetic pathway for the production of sorgoleone in root hair cells [14]. Identification of genes controlling the production of allelochemicals would help in improving cereals for enhancing the release of these chemicals.

The study of genetic mechanism involved in the allelopathic effect of crop plants is a new challenge for biological weed control but very little work has been done till present. Most of the research performed is centered on biochemical analysis and there is extremely limited knowledge on genetic aspect of such characteristics. Especially in sorghum there is no information about the chromosomal regions that trigger the release of allelochemicals. Due to lack of such important information, we have designed our experiment to map the chromosomal regions associated with allelopathic effect of sorghum. We have concluded two new and original hypotheses, (1) Sorgoleone may not be the only phenolic compound responsible for the allelochemicals characteristics in sorghum and other parts of sorghum also produce such chemicals which inhibit the growth of other plants. (2) There might be several QTLs/genes controlling this complex mechanism in sorghum. This study will serve as a pioneer work in the genetic study of sorghum allelopathic effects and will help in marker-assisted selection for

the improvement of allelopathic characteristics in cereals. Our findings will give a base for better understanding of regulatory mechanism of crop allelopathy.

## Materials and methods

### Plant materials

For association mapping, we used our previously developed sorghum diversity research set (SDRS) of 107 accessions (landraces) derived from Asian and African countries [15]. The accessions are characterised geographically into three different groups consisting of (i) East Asia (25 accessions) and Southeast Asia (2 accessions), (ii) South Asia (26 accessions) and Southwest Asia (2 accessions) and (iii) Africa (52 accessions). All plant materials were grown in a glasshouse under natural day-length condition (12L/12D) in 2010 sorghum growing season at agricultural research farm of the University of Tsukuba in Japan (36° N, 139° E). Seeds were treated with fungicide before sowing. Six seeds per accession were sown in round plastic pots (20 cm diameter × 25 cm tall). Regular irrigation, fertilizer and nutrients were applied during the plant growth and insecticide was also used after germination. The plants at six-leaf stage were harvested for bioassay about 45 days after germination.

In linkage mapping, we developed a population from a cross between two sorghum accessions, non-allelopathic Tokibi from Japan (ID; 104JP) and allelopathic Phatsai from Morocco (ID; 33MA) selected from SDRS [15]. The cross was made in year 2008. $F_1$ seeds were sown in 2009 and interbred to produce $F_2$ seeds. During the 2010 growing season, all $F_2$ plants were grown at the distance of 30 cm apart in a field at University of Tsukuba. A total of 134 $F_2$ plants were secured and used for linkage mapping. $F_2$ plants were self-pollinated to produce $F_3$ family lines and subsequently $F_3$ seeds were harvested from each $F_2$ plant. Thirty $F_3$ seeds per $F_2$ plant were sown in pots and leaves were harvested at six-leaf stage about 45 days after germination.

### DNA extraction

For DNA extraction the protocol of Shehzad et al. 2009b [16] was used. Briefly, 40 days old seedlings were selected genomic DNA was isolated from leaf tissues using a modified cetyltri-methylammonium bromide (CTAB) method. The extraction buffer contained 2% (mg/L) CTAB, 50 mM Tris·HCl (pH 8.0), 10 mM EDTA, 0.7 M NaCl, 0.1% SDS, 0.1 mg/ml proteinase K, 2% insoluble polyvinylpyrrolidone (PVP), and 2% 2-mercaptoethanol. Further purification was performed by extraction in chloroform:isoamyl alcohol (24:1 v/v). Then the DNA was precipitated in 2-isopropanol and the precipitate was finally washed in 70% then 90% ethanol. The DNA pellet was dissolved in 50 μl of 1/10 TE buffer containing RNase "A" enzyme, then incubated at 42 °C for an hour. After quantification with V-630Bio (JASCO) spectrophotometer the final DNA concentration of 10 ng/μl was maintained as working dilution.

### Screening SSR markers and genotyping

In our previous study the SDRS was genotyped using 98 SSR markers [16]. A new set of SSRs were selected from Yonemaru et al. [17], which reported more than 5000 newly developed SSR markers from sorghum whole-genome shotgun sequences. In this report, we selected 672 random SSR loci from whole genome of sorghum. After screening with 8 diverse landraces, we selected the best 83 markers (S1 Table) with clear banding pattern. In total, a genotypic data of 181 SSRs were used for association study in SDRS. The parent of $F_2$ lines were also screened with SSRs and finally 152 polymorphic loci were selected for genotyping covering the whole genome of sorghum. Among them, 124 SSRs were from Yonemaru et al. [17] and 28 from other published maps. The PCR reaction and electrophoresis were performed according to the

**Table 1. Correlation among eight parameters studied on lettuce treated with WSE derived from SDRS (lower half) anf F3 (upper half) lines.**

| Traits | Germ25% | Germ50% | Germ75% | Germ100% | RL25% | RL50% | RL75% | RL100% |
|---|---|---|---|---|---|---|---|---|
| **Germ25%** | 1 | 0.012NS | 0.076NS | 0.018NS | 0.033NS | 0.064NS | 0.071NS | 0.091NS |
| **Germ50%** | 0.135NS | 1 | 0.521*** | 0.076NS | 0.230* | 0.223* | 0.103NS | 0.378*** |
| **Germ75%** | 0.097NS | 0.435*** | 1 | 0.428*** | 0.224* | 0.190NS | 0.189NS | 0.359*** |
| **Germ100%** | 0.018NS | 0.075NS | 0.474*** | 1 | 0.366** | 0.444*** | 0.563*** | 0.394*** |
| **RL25%** | 0.033NS | 0.230* | 0.224* | 0.366** | 1 | 0.814*** | 0.704*** | 0.354*** |
| **RL50%** | 0.064NS | 0.223* | 0.190NS | 0.444*** | 0.814*** | 1 | 0.684*** | 0.754*** |
| **RL75%** | 0.071NS | 0.110NS | 0.225* | 0.477*** | 0.704*** | 0.843*** | 1 | 0.518*** |
| **RL100%** | 0.083NS | 0.228* | 0.348** | 0.505*** | 0.633*** | 0.751*** | 0.806*** | 1 |

Germ: Germination; RL: Root Length; SDRS: Sorghum Diversity Research Set.

*** Signficant at $P < 0.001$,

** Significant at $P < 0.01$,

* Significant at $P < 0.05$, NS Non significance.

protocols given in [16]. After screening with molecular markers, genotype "A" was assigned to female parent (Tokibi) allele, "B" for the male parent (Phatsai) allele and "H" for heterozygote.

## Bioassay of allelopathic effect using water-soluble extract (WSE)

The bioassay method reported in Ebana et al. 2001 applied to assess allelopathic effect among sorghum accessions. In this method, water-soluble extracts were prepared form all accessions by grinding freeze dried leaf samples (6-leaf stages) to powder in a mortar and stirred in cold and sterilized distilled water at the rate of 10 ml/g of fresh sample. The mixture was chilled in refrigerator for 2 hours and after stirring, centrifuged at 15, 00 rpm for 15 minutes. The supernatant was collected and diluted with distilled water to prepare solutions of four different concentrations (v/v): 25, 50, 75 and 100%. Lettuce as susceptible to allelochemicals was used as a test plant. Fifty seeds of the lettuce variety Great Lakes 366 were placed on a filter paper (Advantec No. 2) in a 9 cm diameter sterilized plastic petri dishes. Three milliliters of each concentration of crude extract and sterilized distilled water as control were applied to lettuce seeds. Lettuce seeds with no treatment were also kept as control with each treatment of WSE. The petri dishes were sealed and incubated at 25 ˚C in dark for three days in randomized complete block design (RCBD). Soon before the start of germination, the outer filter paper was removed from Petri dishes. The Petri dishes were kept moist consistently by applying distilled water. Data was recorded on germination and root length of 10 randomly selected seeds in 5 replications for each extract and 2 replications for each accession. A total of 250 lettuce seeds were used in data recording for every treatment (concentration) in each replication. Correlation among these traits studied on lettuce is given in Table 1.

## Statistical analysis

Phenotypic values of each accession and $F_3$ lines were represented by the mean values of % germination and root length of lettuce seedlings treated with water-soluble extracts. For one-way analysis of variance (ANOVA) and non-parametric correlation, a statistical program, JMP v.10 [18] was used. Broad-sense heritability ($h^2$) of the traits was also performed using the following formula;

$$h^2 = V_G/V_P$$

where, h2 is broad sense heritability

$$V_G(\text{Genetic Variance}) = \text{Mean Square}(MS)\ \text{Genotypes} - MS(\text{Phenotypes})/\text{Total}$$

$$\text{Genotypes and } V_P(\text{Phenotypic Variance}) = V_G + V_E(\text{Error Variance})$$

A linkage map of SSR markers was constructed using Mapmaker Exp/3.0b software [19]. The mapping function of Kosambi used to convert recombination frequencies were into map distances in cM.

## QTL mapping of $F_3$ lines

The statistical package WinQTLCart 2.0 [20] was used for QTL analysis using the algorithm of composite interval mapping (CIM). For each trait, LOD score threshold was identified using 1000 permutations at P-value = 0.05. We maintained a strict threshold of LOD $\geq$ 2.5 to identify the putative QTLs linked to the traits. The Model 6 of WinQTLCart 2.0 in CIM was used to estimate likelihood of each QTL and its corresponding effects were estimated every 1 cM. A forward–backward stepwise regression was applied to establish the number of significant marker cofactors for background control. A window size of 30 cM was utilized, and consequently, cofactors within 30 cM on either side of the QTL test site were not included in the QTL model. The adjacent QTLs identified for the same trait with non-overlapping intervals on same chromosome were considered as different QTLs. The total phenotypic contribution rate ($R^2$) was calculated as the percent of variation explained by each QTL. A standard nomenclature was applied each QTL with italicized insignia consisting of "*qtl*", one or two digits representing the chromosome number, a hyphen followed by an extra digit if more than one QTL was found on the same chromosome for the same trait, and ending with a trait symbol.

## Population structure and association mapping

The statistical software STRUCTURE version 2.2 [21] was used to perform population structure. After the initial $10^5$ cycles of burn-in period, Markov-chain Monte Carlo (MCMC) sampling was repeated $10^6$ times. Bayesian clustering was applied in population structure analysis with the admixture model. The number of subpopulations ($J$) ranged from 1 to 10 and each run was repeated three times. The optimal number of $J$ was determined on the basis of estimated logarithmic posterior probability of the Bayesian clustering and ad hoc statistic Delta ($\Delta$) $J$. Here, $J = 3$ was selected because it had the largest value of posterior probability and $\Delta J$ among other values (S1 Fig).

A population structure (**Q**) matrix, whose ($i,j$)-th element was expressed as $q_{ij}$, was incorporated into the association mapping model where structure effect was examined. Similarly, a kinship matrix, **K**, was calculated as the allele-sharing rates of the 181 SSR loci as suggested by Zhao et al. [22]. To measure the LD between SSR markers, standard disequilibrium coefficients, D and *r* were used. Where D is the standardized disequilibrium coefficient and *r* represents the correlation between alleles at two loci. For association analysis, the statistical software TASSEL (Trait Analysis by Association, Evolution, and Linkage) ver. 5.0.8 [23] was used and the P-values representing the significance of LD was measured (S2 Fig). The mixed linear model (MLM) was applied to identify QTLs significantly associated with allelopathic characteristics in sorghum. After comparing multiple models of associations, the model that utilized both population structure and kinship (Q+K) was selected. This model identifies P-values based on nominal test of individual makers and then corrected for multiple testing. The P-values obtained were converted into–Log10 (P). The false discovery rate (FDR) of P values was calculated as described by Benjamini & Hochberg [24]. The "BH" method implemented in R

package was used for this purpose. In this method, the *P*-values are first sorted in decreasing order and then they are ranked starting from 1 given to smallest until the last value. The FDR-corrected *P*-value is calculated as *P*-value*(total number of hypotheses tested) / (rank of the *P*-value). The threshold of significance was set as 2.0 value of–$\text{Log}_{10}$ (FDR-corrected *P*).

In the current study, we used regional association mapping to validate 11 major QTLs of germination and root length studied on lettuce using four concentrations of WSE that were identified by the preliminary linkage mapping. As genome-wide association studies results in spurious associations in particular with low number of markers. To control high false positive rates and for validation and of the target major QTLs we applied regional association analysis. The previously selected model of association study i.e. Q+K implemented in MLM method of TASSEL 5.0.8 software was used. The significant marker-trait associations were declared for $P \leq 0.01$.

## Marker localization and homology with known genes

We physically localized the SSR markers that were strongly linked with QTLs in this study by BLAST searches of sequences in http://www.phytozome.net/sorghum (accessed on 15 Nov 2019), http://www.plantgdb.org/SbGDB/ (accessed on 15 Nov 2019), and http://www.gramene.org/ (accessed on 15 Nov 2019). The Genome Data Viewer of NCBI (https://www.ncbi.nlm.nih.gov/genome/gdv/, accessed on 15 Nov 2019) was used to identify loci previously identified as linked to known genes in genome-based sequence information. After identification of maximum matching sequences, Genome Data Viewer was further utilized for searching the database. The sorghum genome database available at http://www.phytozome.net/sorghum was utilized to search the primer sequences when Map Viewer was unable to identify the position. Protein sequences predicted from genes were also used to search using BlastP of NCBI, and finally the homologous sorghum genes were identified by using the sorghum genome in http://www.plantgdb.org/SbGDB/.

## Results

### Assessment of phenotypes

Water-soluble extract (WSE) from each accession of SDRS was tested on lettuce seeds in four concentrations (25%, 50%, 75% and 100%). The allelopathic effect of each accession was calculated based on inhibition of germination and root length of lettuce seeds treated with WSE extracted from it at six-leaf stage. The control (no treatment of WSE) was also used for comparison. Among four concentrations of WSE, wide ranges of variations were observed in 100% (no dilution) of WSE derived from 107 accessions of SDRS as shown in Fig 1a. The frequency distribution of the allelopathic effect was continuous for both traits ranging from 17.5 to 100% for seed germination whereas, 14.1 to 78.8% of the control. Correlation among traits studied in SDRS and $F_{2:3}$ lines shows germination of lettuce seeds treated with 25% WSE had non-significant relationship for all other traits (Table 1). Whereas, root length of lettuce treated with 100% WSE had highly significant correlations with all other traits except germination at 25% WSE in both types of populations. The same way of phenotyping was also performed in $F_2$ derived $F_3$ lines of the cross Tokibi x Phatsai. A wide range of variations were recorded for germination and root length of lettuce seeds treated with 100% WSE derived from $F_3$ lines (Fig 1b). In this case also the frequency distribution was continuous ranging from 5 to 100% and 1 to 79% for root length of lettuce plants treated with WSE relative to control. Germination and root length of lettuce seeds treated with Tokibi were recorded as 95 ± 2.1% and 75 ± 1.4% respectively of the control while for Phatsai, 18 ± 2.1% germination and 14.5 ± 1.4% root length of the control were recorded. Analysis of variance and heritability of traits studied on

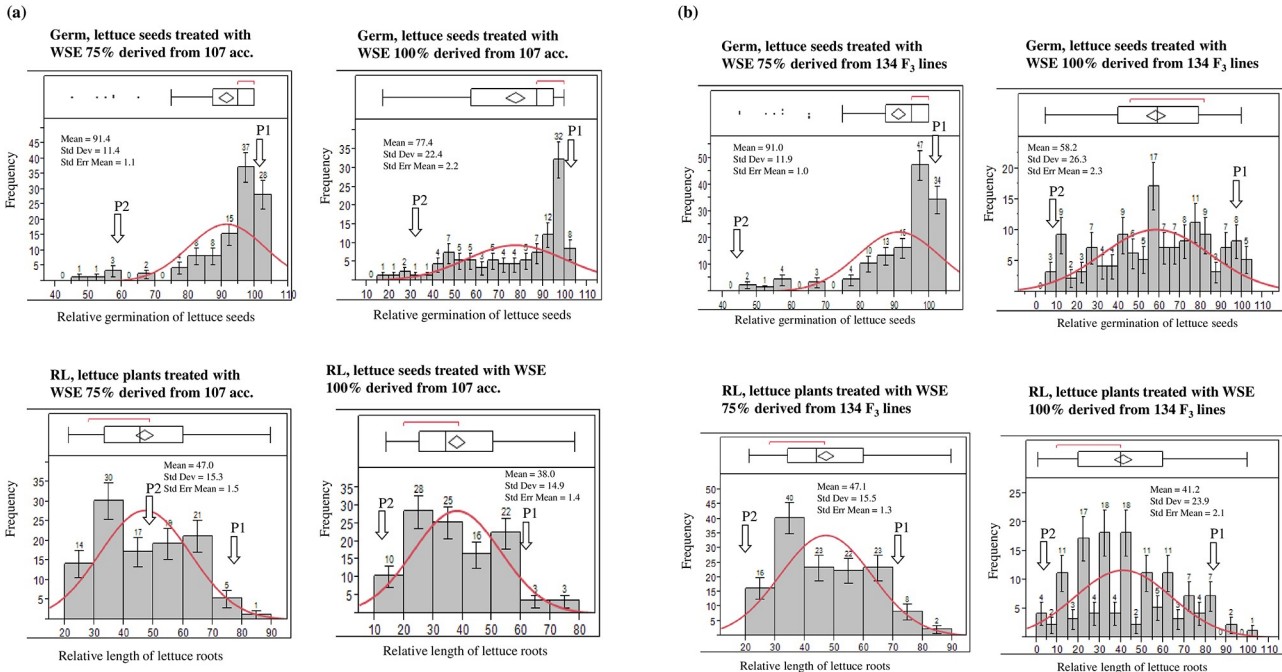

**Fig 1.** (a) Distribution of germination and root length of lettuce treated with water soluble extract (WSE) of 107 sorghum accessions (SDRS) seedlings used in 100% concentration. (b) Distribution of germination and root length of lettuce treated with water soluble extract (WSE) of 134 $F_2$ derived $F_3$ family lines. The red curve is the normal density curve. The top part is Quantile Box Plot (the Outlier Box Plot) and the disconnected points are potential outliers. A red bracket defines the shortest half of the data (the densest region). The results of the first, second, and third run of the outlier identification are displayed in each individual plot from left to right. P1/P2 represents Parent 1 (Tokibi) and Parent 2 (Phatsai), the arrow indicates its trait value where it falls. The term relative represents % germination or root length of the values recorded for control.

$F_{2:3}$ lines showed highly significant variations as well as high heritability for germination (100% WSE) and root length (25%, 50%, 75% and 100%) as shown in Table 2. Germination at 25%, 50% and 100% WSE had non-significant variations and also low heritability values. Based on phenotypic evaluations, lettuce treated with 75% and 100% showed the maximum strength to divide the plant material significantly. So these two concentrations were selected as the most affective ones.

## Construction of linkage map and family-based QTL mapping

The linkage map was constructed from the selected 175 polymorphic markers. A total of 151 SSR loci (S1 Table) were mapped over 10 sorghum chromosomes excepting unlinked 24 SSRs (Fig 2). In this study, 8.2 cM was the average distance between markers whereas, the longest distance was 32.6 cM, and the shortest was 0.5 cM. All of the previously mapped 30 SSRs used among other markers in this study were mapped on the same chromosomes as previously reported (Fig 2).

CIM identified five QTLs responsible for the inhibition of germination of lettuce seeds in $F_{2:3}$. Among them, two QTLs were detected for 25% WSE located on Chr 1 (*qtl1Germ*) and Chr 8 (*qtl8Germ*) (Table 3, Fig 2). The LOD scores were 11.2 and 3.4, whereas $R^2$ were 30.1% and 10.4%, respectively. A single QTL, *qtl2Germ* was found significantly associated with inhibitory effects on germination by 50% WSE with LOD value of 3.5 and $R^2$ of 13.2%. The same QTL (*qtl2Germ*) with flanking markers *Xtxp197* (183.0 cM) and *Xtxp100* (187.4 cM) was also

**Table 2. Analysis of variance (ANOVA), distribution, heritabilty ($h^2$) and genetic advance (R) of seven yield and yield components studied in 134 $F_{2:3}$ lines.**

| Traits | df | Mean squares | Vg[a] | Ve[b] | Vp[c] | F ratio | P > F | Mean | Std Dev | Min | Max | $h^2$ |
|---|---|---|---|---|---|---|---|---|---|---|---|---|
| Germ25% | 133 | 85.2 | 7.1 | 82.1 | 28.5 | 1.7 | 0.049 | 97.7 | 6.3 | 50.0 | 100.0 | 0.24 |
| Error | 134 | 21.5 | | | | | | | | | | |
| Germ50% | 133 | 69.2 | 4.8 | 59.7 | 64.4 | 1.2 | 0.224 | 96.2 | 7.7 | 45.0 | 100.0 | 0.07 |
| Error | 134 | 59.7 | | | | | | | | | | |
| Germ75% | 133 | 260.0 | 84.2 | 91.5 | 175.7 | 2.8 | <0.0001 | 91.2 | 11.4 | 45.0 | 100.0 | 0.47 |
| Error | 134 | 91.5 | | | | | | | | | | |
| Germ100% | 133 | 1026.6 | 467.1 | 92.5 | 559.6 | 11.1 | <0.0001 | 77.2 | 22.9 | 17.5 | 100.0 | 0.83 |
| Error | 134 | 92.5 | | | | | | | | | | |
| RL25% | 133 | 392.0 | 164.5 | 63.0 | 227.5 | 6.2 | <0.0001 | 1.3 | 0.2 | 0.8 | 2.1 | 0.72 |
| Error | 134 | 63.0 | | | | | | | | | | |
| RL50% | 133 | 541.4 | 242.8 | 27.3 | 298.6 | 9.7 | <0.0001 | 1.0 | 0.3 | 0.5 | 1.8 | 0.81 |
| Error | 134 | 55.8 | | | | | | | | | | |
| RL75% | 133 | 470.4 | 221.6 | 27.3 | 248.8 | 17.2 | <0.0001 | 0.8 | 0.2 | 0.3 | 1.4 | 0.89 |
| Error | 134 | 27.3 | | | | | | | | | | |
| RL100% | 133 | 437.3 | 185.7 | 65.9 | 251.6 | 6.6 | <0.0001 | 0.6 | 0.3 | 0.3 | 1.4 | 0.74 |
| Error | 134 | 65.9 | | | | | | | | | | |

Genotypic mean variance is classified into genetic varaince (Vg), error varaince (Ve) and total phenotypic varaince (Vp).

found strongly associated with the trait treated with 75% WSE with highest LOD score explaining 35.6% of total phenotypic variance. Another QTL with name *qtl5Germ* was also related to this trait studied on 75% WSE with 6.1 of LOD score and 16.0% of total phenotypic variance ($R^2$). Similarly, a single QTL (*qtl3Germ*) significantly inhibited germination in 100% WSE, indicating LOD score of 4.3 and $R^2$ of 12.4%.

A single QTL, *qtl4RL* was identified to inhibit root length of lettuce seedlings treated with 25% WSE derived from $F_{2:3}$ lines with LOD score of 2.9 and controlling 9.9% of total phenotypic variation (Table 3, Fig 2). In case of 50% WSE used, maximum number of QTLs (five) were identified as strongly associated with root inhibitory effect with LOD score values ranging from 2.7 to 5.3. $R^2$ ranged from 10.5% to 22.1%. These include *qtl2RL*, *qtl6RL*, *qtl7RL*, *qtl8RL* and *qtl10RL* located on Chrs 2, 6, 7, 8 and 10, respectively.

For 75% WSE, two QTLs were identified for root length inhibition, on Chr 4 (*qtl4RL*) and Chr 10 (*qtl10RL*), with LOD scores of 4.6 and 9.8, respectively and R2 values of 16.1% and 34.5%, respectively. Similarly for 100% WSE, two QTLs were detected as significantly controlling inhibitory action of WSE on root length of lettuce. These include *qtl1RL* with LOD score of 2.4 and $R^2$ value of 11.5% on Chr 1 and another QTL *qtl10RL* with LOD score of 3.7 and $R^2$ value of 14.3%. Among these QTLs, *qtl10RL* with flanking markers *Xtxp270* and *SB5329* at position 32.6–35.5 cM was commonly associated with the root length inhibition of lettuce treated with 50%, 75% and 100% WSE.

## Population structure and genome-wide association study

In this study, 181 SSR markers were used to genotype 107 accessions of SDRS covering the whole genome of sorghum. The number of subpopulations (*J*) was tested from two to ten, each repeated three times. The posterior probability and ad hoc statistic values of *J* = 3 were the largest among other subpopulations (S1 Fig). Therefore, we chose *J* = 3 and obtained estimates of $q_{ij}$ for the proportion of *i*'s genome that originated from population *j*. The first subpopulation

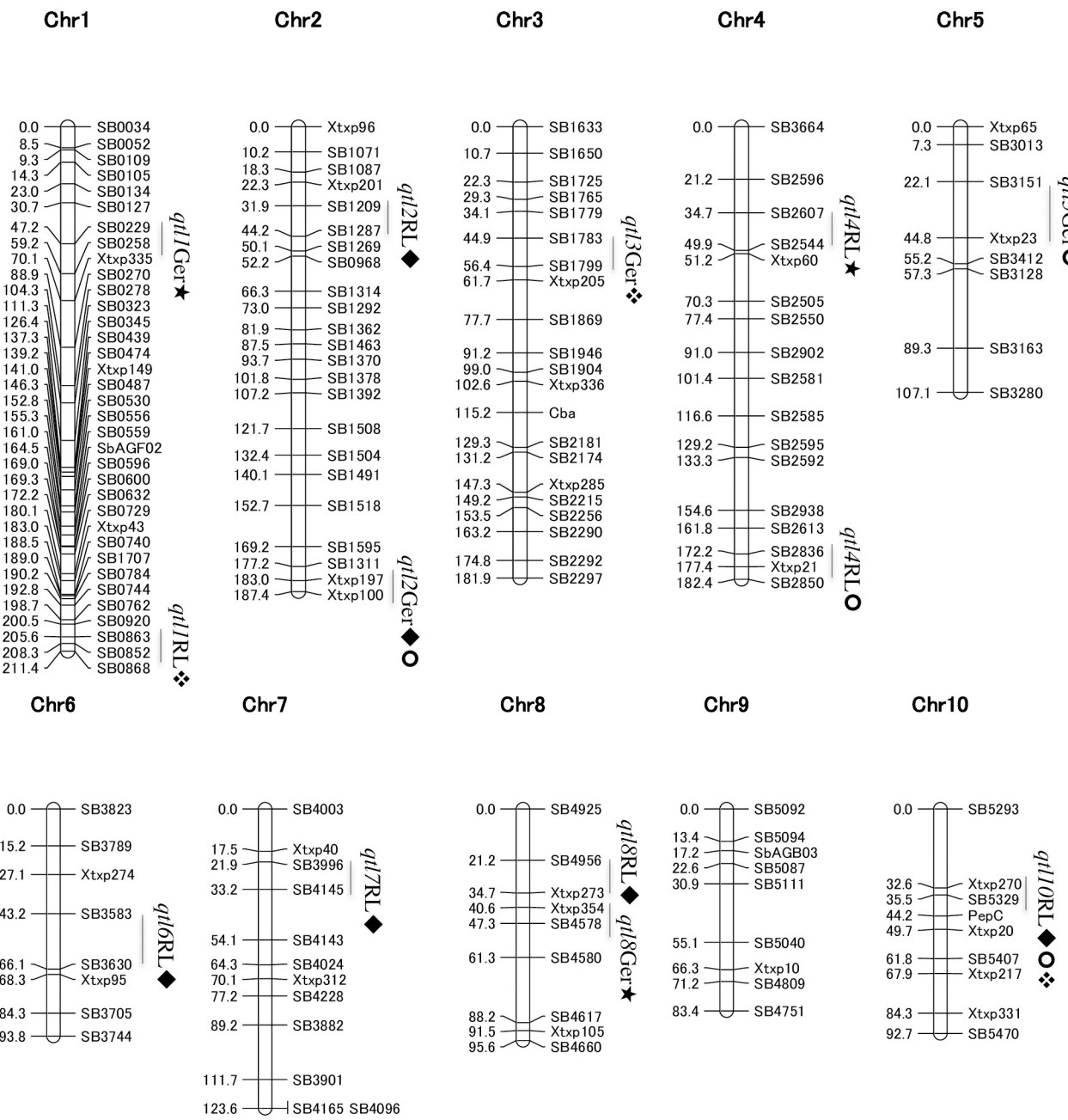

**Fig 2. Association analysis of 181 SSR markers and eight traits studied on germination and root length of lattice plants using water soluble extracts from 107 sorghum accessions (SDRS).**

(*J* = 1) was composed of 33 accessions: 27 from Africa and 6 from Asia. The second subpopulation (*J* = 2) was the largest group that comprised 39 accessions from East and South Asian countries. The third subpopulation (*J* = 3) had 35 sorghum accessions derived from African and Asian origin. A low to medium range of LD among SSR loci were observed in this study. To control false positives, both population structure (**Q**) and kinship (**K**) were implemented in MLM model of association analysis that is generally called a complete model or Q+K model. Using this model, two significant loci were associated with germination of lettuce seeds treated

**Table 3. QTLs identified by linkage analysis using composite interval mapping (CIM) model for the effects of WSE from 134 F3 lines on germination and root length growth of lettuce plant.**

| Trait | WSEa | QTL | Chromosome | Marker/Interval | Position (cM) | Physical Position (Start-End) of Markers | F3 | | | |
|---|---|---|---|---|---|---|---|---|---|---|
| | | | | | | | Effects | | $R2$ (%) | LOD |
| | | | | | | | Add | Dom | | |
| Germination | 25% | qtl1Germ | 1 | SB0229-SB0258 | 47.2–59.2 | (157142..157290)-(11943960..11944153) | 1.4 | -0.7 | 30.1 | 11.2 |
| | | qtl8Germ | 8 | Xtxp354-SB4578 | 40.6–47.3 | (48122788..48122851)-(51583239–51583426) | 7.1 | -7.5 | 10.4 | 3.4 |
| | 50% | qtl2Germ | 2 | Xtxp197-Xtxp100 | 183.0–187.4 | (1449013..1449076)-(69473471..69473534) | -5.9 | -6.7 | 13.2 | 3.5 |
| | 75% | qtl2Germ | 2 | Xtxp197-Xtxp100 | 183.0–187.4 | (1449013..1449076)-(69473471..69473534) | -7.4 | -0.5 | 35.6 | 12.4 |
| | | qtl5Germ | 5 | SB3151-Xtxp23 | 22.1–44.8 | (9589292..9589470)-(54447105..54447168) | -7.8 | -0.6 | 16.0 | 6.1 |
| | 100% | qtl3Germ | 3 | SB1783-SB1799 | 44.9–56.4 | (7055966..7056239)-(6841185..6841450) | 7.3 | -7.1 | 12.4 | 4.3 |
| Root Length | 25% | qtl4RL | 4 | SB2607-SB2544 | 34.7–49.9 | (17481687..17481840)-(7368441..7368710) | -2.6 | -2.0 | 9.9 | 2.9 |
| | 50% | qtl2RL | 2 | SB1209-SB1287 | 31.9–44.2 | (57080999..57081201)-(61259880..61260075) | -7.0 | -8.7 | 22.1 | 5.3 |
| | | qtl6RL | 6 | SB3583-SB3630 | 43.2–66.1 | (49670815..49670986)-(52813876..52814028) | 9.2 | -7.5 | 12.4 | 2.7 |
| | | qtl7RL | 7 | SB3996-SB4145 | 21.9–33.2 | (7788962..7789259)-(58412340..58412533) | 0.4 | -8.7 | 12.2 | 2.5 |
| | | qtl8RL | 8 | SB4956-Xtxp273 | 21.2–34.7 | (51161240..51161383)-(157079..157142) | -6.0 | -6.4 | 17.0 | 3.1 |
| | | qtl10RL | 10 | Xtxp270-SB5329 | 32.6–35.5 | (11037040..11037103)-(41184605..41184836) | -2.1 | -6.1 | 10.5 | 2.8 |
| | 75% | qtl4RL | 4 | SB2836-Xtxp21 | 172.2–177.4 | (62144483..62144661)-(67894269..67894331) | -7.1 | -8.0 | 16.1 | 4.6 |
| | | qtl10RL | 10 | Xtxp270-SB5329 | 32.6–35.5 | (11037040..11037103)-(41184605..41184836) | -0.9 | -7.6 | 34.5 | 9.8 |
| | 100% | qtl1RL | 1 | SB0863-SB0852 | 205.6–208.3 | (71693225..71693516)-(70830608..70830842) | -7.5 | -2.4 | 11.5 | 2.4 |
| | | qtl10RL | 10 | Xtxp270-SB5329 | 32.6–35.5 | (11037040..11037103)-(41184605..41184836) | -8.5 | -8.4 | 14.3 | 3.7 |

[a]; Water soluble extract of sorghum accessions used in four different concentrations of 25%, 50%, 75% and 100%.

with 25% of WSE at $P \leq 0.001$ (Table 4, Fig 3). One of them was located on chromosome (Chr) 1 (*Xtxp335*) and was named *qtl1Germ* while the other was located on Chr 8 (*qtl8Germ*) linked to *Xtxp47*. The–Log$_{10}$ (*P*) of these QTLs were 2.2 and 2.1 while explained phenotypic variance ($R^2$) were 17.1 and 9.2, respectively. In treatment of 75% WSE, a single locus (*Xtxp100*) was strongly associated with germination located on Chr 2. This QTL (*qtl2Germ*) had the maximum–Log$_{10}$ (*P*) value of 4.2 and explaining 14.8% of the phenotypic variance.

The maximum QTLs *i.e.* four were resolved for the inhibitory effect on root length treated with 50% of WSE on chromosomes 4 (*qtl4RL*), 7 (*qtl7RL*), 8 (*qtl8RL*) and 10 (*qtl10RL*). The

**Table 4. QTLs identified by association mapping using Q+K model for the effects of WSE from SDRS on germination and root length growth of lettuce plant.**

| Trait | WSE[a] | QTL | Chromosome | Marker | Start Position (bp) | End Position (bp) | -Log10(P) | % R$^2$ |
|---|---|---|---|---|---|---|---|---|
| Germination | 25% | qtl1Germ | 1 | Xtxp335 | 55750529 | 55750592 | 2.2 | 17.1 |
| | | qtl8Germ | 8 | Xtxp47 | 2912606 | 2912668 | 2.1 | 9.2 |
| | 75% | qtl2Germ | 2 | Xtxp100 | 69473471 | 69473534 | 4.2 | 14.8 |
| Root Length | 50% | qtl4RL | 4 | SB2836 | 62144483 | 62144661 | 2.7 | 21.7 |
| | | qtl7RL | 7 | SB4003 | 8643296 | 8643486 | 2.0 | 6.9 |
| | | qtl8RL | 8 | Xtxp273 | 157079 | 157142 | 2.1 | 13.0 |
| | | qtl10RL | 10 | Xtxp270 | 11037040 | 11037103 | 2.0 | 23.6 |
| | 75% | qtl10RL | 10 | Xtxp270 | 11037040 | 11037103 | 3.6 | 27.3 |
| | 100% | qtl1RL | 1 | SB0863 | 71693225 | 71693516 | 2.2 | 22.6 |
| | | qtl10RL | 10 | Xtxp270 | 11037040 | 11037103 | 2.1 | 22.5 |

[a]; Water soluble extract of sorghum accessions used in four different concentrations of 25%, 50%, 75% and 100%.

[b]; -Log$_{10}$ of *P*-values determined for Q+K model with 2.0 as threshold value for strong association.

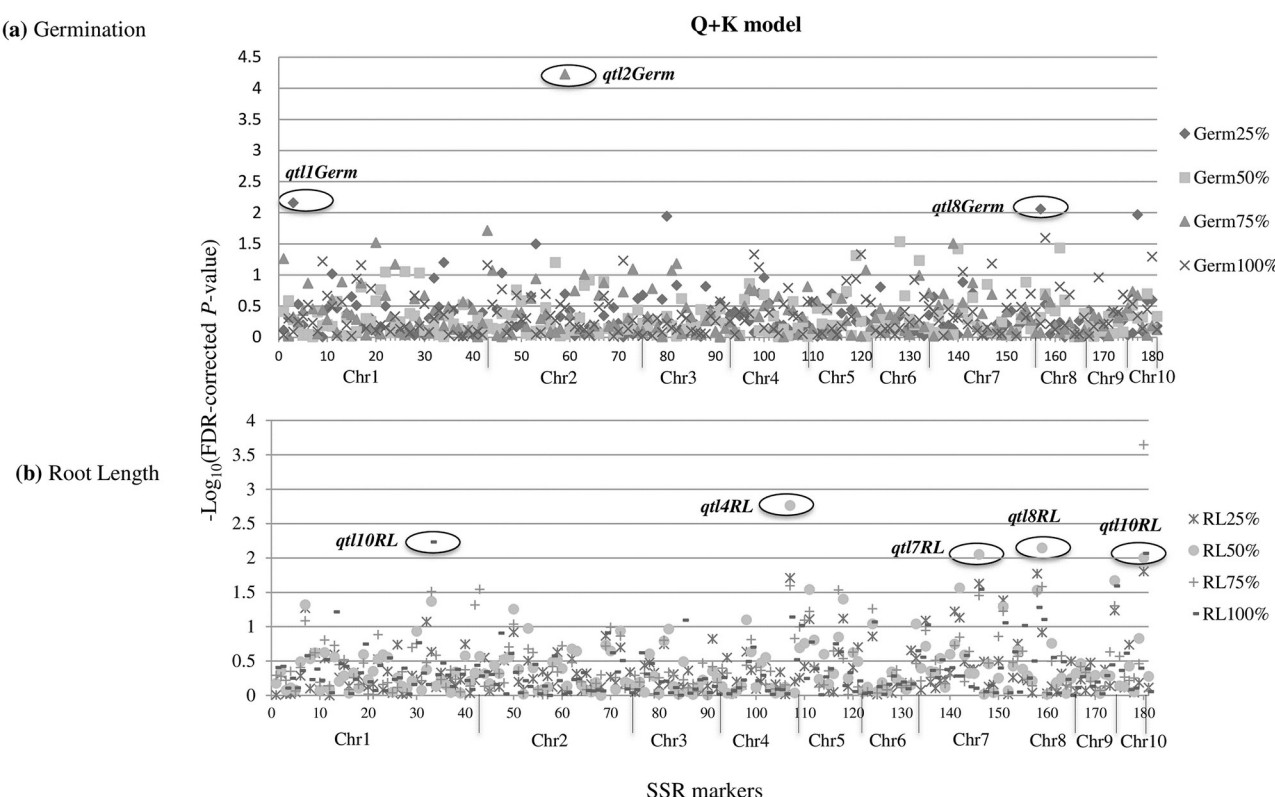

**Fig 3. Linkage map generated using 151 SSR markers on 134 F₂ population.** QTL analysis was performed on eight allelopathy related traits studied on germination and root length of lettuce plants exposed to water soluble extracts (WSE) from F₃ family lines. WSE was used in four different concentrations i.e. 25% (★), 50% (◆), 75% (●) and 100% (❖).

values of–$\mathrm{Log}_{10}$ (P) were recorded as 2.7 ($R^2$ = 21.7), 2.0 ($R^2$ = 6.9), 2.1 ($R^2$ = 13.0) and 2.0 ($R^2$ = 23.6), respectively. A single locus *Xtxp270* was found significantly associated with root length inhibition by 75% WSE with–$\mathrm{Log}_{10}$ (P) value of 3.6 and explaining 22.5% of total phenotypic variance. Similarly two loci (*SB0863* and *Xtxp270*) showed highly significance at $P \leq 0.001$ with the root length inhibition trait (RL) by 100% WSE. These QTLs were located on chromosomes 1 (*qtl1RL*) and 10 (*qtl10RL*) with–$\mathrm{Log}_{10}$ (P) values of 2.2 ($R^2$ = 22.6) and 2.1 ($R^2$ = 22.5), respectively (Table 4, Fig 3). No QTL was detected for germination of lettuce seeds treated with 50 and 100% WSE.

## Regional association mapping

The corresponding genomic regions of 13 major unique QTLs were identified by the alignment between the primer sequences of tightly linked SSR markers. Considering the population structure and family relatedness within the population, the association analysis was conducted with a mixed linear model (MLM) by TASSEL ver. 5.0.8 using the 107 landraces (SDRS) and 21 QTL-linked SSR loci in the target regions. Notably, 14 out of 21 loci showed lower–$\mathrm{Log}_{10}$ of P-values than the threshold of 2.5 whereas, seven loci had significant values. At strict threshold of 3.0, only four loci were highly significantly associated with allelopathic characteristics of SDRS (Fig 4). Among them, two SSRs loci, *Xtxp100* (Chr 2) and *SB3630* (Chr 6) were associated with germination at 75% WSE treatment having -$\mathrm{Log}_{10}$ (P) of 4.2 ($R^2$ = 14.8) and 3.6 ($R^2$ = 12.3), respectively. The locus *SB0863* (Chr 1) showed strong association (-$\mathrm{Log}_{10}$

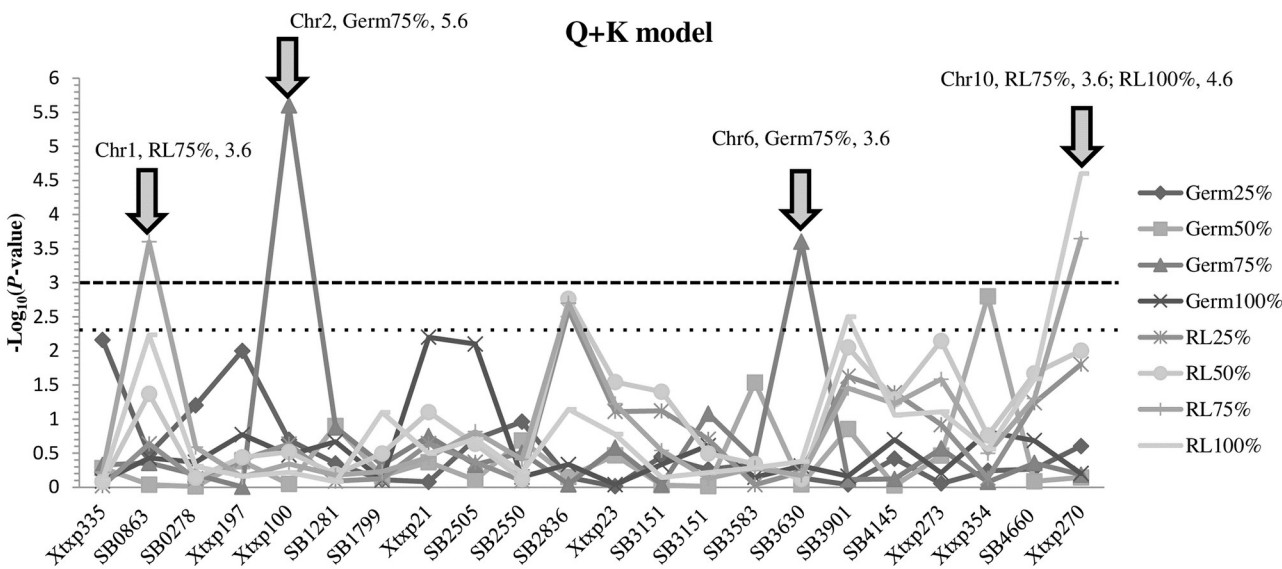

**Fig 4. Targeted association of 22 QTL-SSR markers and eight traits studied on germination and root length of lattice plants using water soluble extracts from 107 sorghum accessions (SDRS).**

$(P) = 3.6$, $R^2 = 11.8$) for root length at treatment of 75% WSE. Another QTL, *Xtxp27* (Chr 10) also had strong association for root length of lettuce treated with WSE of sorghum accessions at 75% (-Log$_{10}$ $P$-value = 3.6, $R^2 = 12.5$) and 100% (Log$_{10}$ $P$-value = 4.6, $R^2 = 21.2$) as shown in Fig 4.

## Physical co-localization with known genes

We examined the position of each QTL identified here with previously known genes by physically localizing the QTL markers on sorghum chromosomes. Several candidate genes were identified in the proximal QTL regions of this study. The locus *SB0852*, which flanks *qtl1RL*, is located on Chr 1 at 70817033 bp, is reported as the location of protein-coding gene *SB01G047730* in a previous study. This gene is predicted to function in protein and nucleic acid binding processes. Another marker locus associated with *qtl1Germ* on Chr 1, *Xtxp335* (55750572 bp) is located within a candidate gene *SB01G032850*, which is predicted to play a role in the light reactions of photosynthesis.

Similarly, *Xtxp273* located at 19929776 bp identified by association analysis on Chr 8 (*qtl8RL*) contained within a candidate gene coding fascidin-like arabinogalactin protein 7 in *Brassica rapa*. A locus *Xtxp354* identified in family-based linkage analysis flanking *qtl8Germ* located on Chr 8 at 48122951 bp and another locus *Xtxp100* (Chr2: 269473512 bp) found significant in both association and linkage mapping encodes reverse transcriptase RVT-2 superfamily gene. This gene can be found in maize, switch grass, sorghum, grape wine and other plants. Marker locus *SB0229* (Chr 1: 10155578) of the QTL *qtl1* Germ found in family-based linkage analysis encodes helitron helicase-like that is recognized eukaryotic transposon predicted to amplify by roling-circle mechanism. Another locus *SB4145* on Chr 7 (58359565 bp) flank with a QTL *qtl7RL* inhibiting root length of lettuce plant contained within a gene (Sb07g023510) that has 25 orthologs in different plant species. Similarly one flanking marker of *qtl10RL* (*SB5329*) on Chr 10 at 41202394 bp, is associated with a candidate gene

Sb10g019340, which encodes a known protein with transferase activity. This gene has 18 orthologs in other plant species.

## Discussion

Allelopathy is an ecological term, derived from two Greek words "Allelon" means mutual and "Pathos", means harm [25]. Allelopathy plays important role in plant to plant interaction and its surroundings through the production of chemical compounds (allelochemicals) that are released into the environment and effecting the whole agroecosystem. Previously, sorghum allelopathy against weeds has been extensively studied for biological weed control. Sorghum residues release a number of allelochemicals such as sorgoleone, dhurrin and a number of secondary metabolites that bring about weed suppression. Sorghum has a great potential for allelopathic crop to control parasitic weeds, especially *Striga* and *Orobanche* species which pose a serious threat to agriculture as they are difficult to manage.

Several strategies have been adopted to control weeds including the use of sorghum extracts, use of sorghum residues as mulch and cover crops or soil incorporation, and also the use of sorghum in crop rotations [26, 27, 10]. Until recent years, a wide range of sorghum allelochemicals have been isolated and characterized from shoots, roots as well as root exudates. Among those chemcals, numerous phenolics, a cyanogenic glycoside (dhurrin), and a hydrophobic p-ben- zoquinone (sorgoleone) have been reported in several studies. In terms of mode of action and specificity, sorgoleone has been widely investigated which is continuously released by living root hairs. The study of genetic factors involved in sorghum allelopathy has not been given much attention except SOR1, a gene associated with the production of sorgoleone [5]. A differential expression analysis and real-time PCR revealed that this gene is expressed in the roots of sorghum but not in shoots. Here we report the genetic analysis of phytotoxins produced in sorghum parts other than sorgoleone in roots.

In this study two different approaches of QTL mapping, LD based mapping and linkage mapping, has been used for the first time to study the genetics of allelopathy in sorghum. LD mapping approach identified three QTLs associated with inhibiting germination and seven QTLs with root length inhibition effect. Whereas, linkage mapping detected total of 16 QTLs including six for inhibition effect of WSE on germination and ten for inhibition effect on root length of lettuce plants.

The association analysis was performed using SDRS assessed with 181 SSR molecular markers. The linkage map reported here, is composed of 151 mapped SSR markers covering all number of sorghum chromosomes. An additional 24 markers were also tested but could not be mapped due to segregation distortions. The SSRs reported here were developed from shotgun sequences of the whole sorghum genome [17]. Similarly, 30 previously mapped SSRs reported by Bhattramakki et al. [28] were also mapped and showed consistency in the pattern of recombination with other markers. All of the 30 loci were mapped to the same chromosomes as previously reported, which shows the accuracy of locating the markers in this new map. Most of the other new loci selected from Yonemaru et al. [17] were mapped to the same chromosomes as previously reported, with few exceptions (ESM1). *SB1707* was mapped on Chr 1 but originally selected from genome sequence of Chr 3. Two other loci, *SB3664* and *SB2613* were mapped on Chr 4 in this study but were localized on Chr 6 and Chr 1, respectively in previous report [17]. Likewise, in Yonemaru et al. 2009 [17] the two markers *SB4925* and *SB4956* are reported on Chr 6 but mapped differently in our study. As these discrepancies are common due to sampling variation or the mapping of paralogous loci (i.e., loci arising from gene duplication). Most of the molecular markers in this study were mapped to the same chromosomes as reported earlier by linkage analysis or physical mapping. This supports the

accuracy and reliability of the linkage map developed here. Some of the markers mapped in this population has also been used in another study by Shehzad and Okuno [29] and showed no discrepancies in mapping these loci on chromosomes.

We used four different concentrations of WSE, affecting the germination and root growth of lettuce plant at different levels. This is due to the contents of allelochemicals in lower to higher concentrations of WSE. Previously, Agarwal et al. [30], Iqbal et al. [31], Fateh et al. [32] and Shang & Xu [33] also reported an increase in phytotoxicity of allelochemicals with increasing concentration. After statistical analyses, large variations were observed in the traits studied on lettuce treated with 75% and 100% WSE concentrations in comparison with 25% and 50% concentrations. This shows 75% and 100% concentration of WSE is more appropriate to study allelopathic characteristics in sorghum.

The two QTL mapping approaches identified several similar and unique sets of QTLs as significantly associated with the allelopathic characteristics in sorghum. This agreement between the methods supports the efficiency and reliability of our findings. The genomic sequence of *SOR1* was physically mapped on Chr 5 (216857–218904 bp). In this study, no QTL was located in the region of where *SOR1* is localized, suggesting that water-insoluble sorgoleone has different chemical properties than the allelochemicals present in water soluble extract. The LD mapping found nine QTLs (three for inhibiting germination and six for root length of lettuce) and linkage study identified total of 17 QTLs (seven for inhibiting germination and 10 for root length of lettuce). These results show the presence of several allelochemicals and mutli-genic nature of allelopathic traits in sorghum. In other crops, such as rice and wheat, allelopathic characteristics associated with several chromosomal regions have been reported, suggesting the presence of several allelochemicals [34–38].

In total, LD-based association mapping identified ten QTLs, including three for germination and seven for root length inhibition. Among them, *qtl10RL* (*Xtxp270*) on Chr 10 was commonly associated with root inhibition at 50%, 75% and 100% WSE. While no QTL was identified for inhibiting germination by 50% and 100% concentrations of WSE and root length by WSE used in 25% concentration. The linkage mapping detected 16 QTLs in $F_{2:3}$ lines including six for germination inhibition and ten QTLs for root length inhibition. Here, *qtl10RL* (*Xtxp270–SB5329*) was also found significantly controlling allelopathic effect on root length of lettuce by WSE used in 50%, 75% and 100% concentrations. We could not find any significant QTL for allelopathic effect of WSE used in 50% and 100%, whereas QTL for root length inhibition with 25% WSE. The QTLs detected in SDRS by association analysis were also mapped in similar positions in $F_{2:3}$ family lines by linkage analysis along with other unique QTLs. This shows the accuracy for our methodologies and confirming the stability of QTLs mentioned in this report.

Some of the QTLs identified here, appear to correspond to previously reported QTLs and genes for other related traits. In sorghum, *dwarf3* (*dw3*) is one of four major dwarfing genes, has been cloned and sequenced by Multani et al. [39]. This gene is an ortholog of *brachytic2* (*br2*) in maize and in sorghum it is mapped on Chr 7. In this study, we also identified a major QTL for root length inhibition effect (*qtl7RL*) on Chr 7 in both approaches of QTL analysis. In LD mapping, SB4003 was found significant with–$Log_{10}$ (*P*-value) as 2.0 (Table 3) whereas, linkage mapping identified this QTL between SSR markers *SB3996* (21.9 cM) and *SB4145* (33.2 cM) (Table 4), which is the same location as *dw3*. Similarly, *qtl6RL* with flanking markers *SB3583* (43.2 bp) and *SB3630* (66.1 bp) falls in the location on Chr 6 where $Ma_1$ gene of sorghum maturity is located. This is the major repressor of sorghum flowering under longer daylength, which encodes a protein (PRR37) modulating flowering time in sorghum [40]. The *qtl10RL* identified in this study (Figs 2, 3 and 4) is in similar genomic regions as the previously identified one of the stay-green QTL regions, *StgG* [41]. The relationship among allelopathic

characteristics and these traits need to be tested and further to establish the nature of allelopathy and the effects of these QTLs.

Both approaches identified different QTLs for allelopathic effect on seed germination and root length of lettuce. This shows two different mechanisms are involved in controlling allelopathic effects at two stages. There might be different sets of allelochemicals affecting plants at different part and/or growth stage. A single common co-localized major QTL *qtl10RL* on Chr 10 between markers *Xtxp270* (32.6 bp) and *SB5329* (35.5 bp) was detected for inhibition of root length by WSE used in 50%, 75% and 100% concentrations in both approaches of QTL mapping. This shows the importance of this region involved in allelochemical characteristics in sorghum.

Regional association mapping helped in controlling spurious association that commonly occurs in genome-wide association analysis. Targeted or regional association mapping using MLM model could significantly control false positive rates thus resulting in more authentic detection of QTLs. In this study, regional association by 21 QTL loci identified total of four major QTLs out of which two were associated with germination and two with root length treated with WSE from sorghum landraces (Fig 4). Among them, a QTL on Chr 2 was identified for germination of lettuce treated with 75% WSE which is same as detected by linkage analysis of $F_{2:3}$ population as well as genome wide association mapping. Another unique QTL for the same trait was identified on Chr 6 ($-Log_{10} P = 3.6$) that has not been shown in linkage and whole genome association mapping. We also detected two QTLs for root length of lettuce on Chr 1 (75% WSE) and Chr 10 (75%, 100 WSE), respectively (Fig 4). Both of these QTLs were co-localized by family based linkage and LD-based whole genome based association analysis.

Further study is required to finely dissect these target regions that will yield towards positional cloning of candidate gene(s). Chemical assessments are also essential for profiling allelochemicals that attribute allelochemical characteristics to sorghum. The results show that sorgoleone is not only phytotoxin existing in sorghum but several phenolic compounds are also involved in water-soluble extracts. Similarly *SOR1* may not be the only gene involved in this phenomenon rather several QTLs/genes are responsible for allelopathy in sorghum. The major QTLs identified in this study could be used for fine mapping and further isolation of genes based on positional or map based cloning approach.

## Supporting information

**S1 Fig. Structure analysis using burn-in length of $10^5$ and MCMC cycles of $10^6$ classified the population into three sub-populations (*J*), where *J* is varied from 1 to 9.** Circles represent independent runs for each value of J. Each run for *J* was repeated three times.
(PPTX)

**S2 Fig. Linkage disequilibrium (LD) plot generated by 181 SSR loci.** Each cell represents the relationship between two markers with color codes showing the level of significance.
(PPTX)

**S1 Table. List of newly added 83 SSR markers used in genotyping of sorghum diversity research set (SDRS).**
(XLSX)

## Author Contributions

**Conceptualization:** Tariq Shehzad, Kazutoshi Okuno.

**Data curation:** Tariq Shehzad.

**Formal analysis:** Tariq Shehzad.

**Funding acquisition:** Kazutoshi Okuno.

**Investigation:** Tariq Shehzad.

**Methodology:** Tariq Shehzad.

**Project administration:** Tariq Shehzad.

**Resources:** Kazutoshi Okuno.

**Validation:** Tariq Shehzad.

**Writing – original draft:** Tariq Shehzad.

**Writing – review & editing:** Tariq Shehzad, Kazutoshi Okuno.

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
