## [Decision Letter · Decision Letter 0]

13 Feb 2020

PONE-D-19-28312

Genetic analysis of QTLs controlling allelopathic characteristics in sorghum

PLOS ONE

Dear Dr. Shehzad,

Thank you for submitting your manuscript to PLOS ONE. After careful consideration, we feel that it has merit but does not fully meet PLOS ONE’s publication criteria as it currently stands. Therefore, we invite you to submit a revised version of the manuscript that addresses the points raised during the review process.

We would appreciate receiving your revised manuscript by Mar 29 2020 11:59PM. To enhance the reproducibility of your results, we recommend that if applicable you deposit your laboratory protocols in protocols.io, where a protocol can be assigned its own identifier (DOI) such that it can be cited independently in the future. For instructions see: http://journals.plos.org/plosone/s/submission-guidelines#loc-laboratory-protocols

We look forward to receiving your revised manuscript.

Kind regards,

Craig Eliot Coleman, PhD

Academic Editor

PLOS ONE

Journal Requirements:

1. Please amend the subsection category “[FOR JOURNAL STAFF USE ONLY]” for your manuscript. Unfortunately, this is not a valid category. At this time, please choose one or more subsections that best represent the topic(s) of your study.

4. Please upload a new copy of Figures 1-4 as the detail is not clear. Please follow the link for more information: http://blogs.PLOS.org/everyone/2011/05/10/how-to-check-your-manuscript-image-quality-in-editorial-manager/

Reviewers' comments:

Reviewer's Responses to Questions

**Comments to the Author**

1. Is the manuscript technically sound, and do the data support the conclusions?

Reviewer #1: Yes

Reviewer #2: Yes

2. Has the statistical analysis been performed appropriately and rigorously? 

Reviewer #1: Yes

Reviewer #2: Yes

3. Have the authors made all data underlying the findings in their manuscript fully available?

Reviewer #1: No

Reviewer #2: Yes

4. Is the manuscript presented in an intelligible fashion and written in standard English?

Reviewer #1: Yes

Reviewer #2: No

5. Review Comments to the Author

Reviewer #1: The authors used two sorghum populations, i.e. SDRS and F2:3, to study the inhibitory effect of sorghum WSE on the seed germination and on the root length of lettuce seedlings. They used four concentrations of the WSE (25%, 50%, 75% and 100%). The authors identified several QTL in the F2:3 and in the association mapping populations. The work is designed in a proper way, however, I have the following comments;

• I couldn't find any figures attached to the manuscript that could have helped me to better understand the experimental design.

• Please, add how the heritability was calculated. Was the H2 for germ50% 0.07?

• Please, remove the last paragraph of the "Bioassay of allelopathic effect using water-soluble extract (WSE)" as it was explained again in the next paragraph "Statistical analysis".

• For QTL mapping, was threshold 2.5 chosen based on 1000 permutation or as an arbitrary threshold?

• I couldn't find any figure or data presenting the map of the F3 or the association panel. Please, provide this data.

• In such low density map, I would recommend using window size larger than 10 cM, e.g. 30 cM, to consider the following sentence correct "The adjacent QTLs identified for the same trait with non-overlapping intervals on same chromosome were considered as different QTLs". In addition, one can consider two adjacent qtl as different ones only when their effect is different, e.g. one has an effect from A and the other from B.

• Please change this title “Linkage disequilibrium (LD)-based association mapping” to “Population structure and association mapping” or add two subtitles “Population structure” and “Association mapping”

• “the P-values representing the significance of LD was measured”, how was P value calculated? What is threshold of 2.5 and strict threshold =3 ? Please explain. Have you used an arbitrary threshold or Bonferroni correction? I believe a P value of 2.5 is relatively low and will increase the false positives.

• Last paragraph of Linkage disequilibrium (LD)-based association mapping, “As genome-wide association studies results in spurious associations in particular with low number of markers.” Not clear what Authors want to say.

• In Marker localization and homology with known genes “The Map Viewer of NCBI website (http://www.ncbi.nlm.nih.gov/mapview/) was used to identify loci previously identified as linked to known genes in genome-based sequence information.” I don’t understand whether the author did that to know if their SSR were previously mapped, which could be done from previously published articles, or they did that to just know what is known. Please, clarify on that.

• In “Construction of linkage map……”, please, move the following sentence to discussion “These findings confirms the consistency and accuracy of the linkage map reported here.

• The discussion part, in general, needs to be restructured to avoid repeating sentences and contents. Below are some suggestions,

• when comparing the discrepancy in locating some SSR, e.g. SB3664 and SB2613, please give references after "in previous report".

• The paragraph that starts with "We used two approaches and two different types of populations ……………..", it reads like a review not discussion. Can be deleted or modified.

• "We used four different concentrations of WSE, ……………...." and then "This confirms...…....". It isn't clear what confirms what.

• It will be more coherent to gather all the discussion related to QTL and LD mapping in one paragraph, as now in several places I find the same conclusion, i.e. the suitability of the 2 approaches to map significant QTL ....."

• Why discussing SOR1? Please take the space needed to elaborate on why specifically the authors discussed this gene

• "A single common co-localized major QTL qtl10RL on Chr 10 between markers Xtxp270 (32.6 bp) and SB5329 (35.5 bp) was detected for inhibition of root length by WSE used in 50%, 75% and 100% concentrations in both approaches of QTL mapping". The content of this sentence was mentioned earlier in the discussion. Please, remove this sentence and all other repeated sentences.

• The author should carefully interpret the results of the "regional mapping" and the physical co-location of with known genes as the QTL interval is normally large and contain several genes, especially with this low dense map. I would rather call them candidate genes

Reviewer #2: Allelopathic characteristics of sorghum has been studied at large, however, the reports of QTL for allelopathic characteristics were not reported. In this line, the study “Genetic analysis of QTLs controlling allelopathic characteristics in sorghum” has a merit. Authors did very good job by conducting experiment in optimal way and results are also very interesting however authors need to focus on the results of the study. I propose to recommend for publication in PLOSONE with few suggestions to improve the manuscript.

Major revision

1. The phytotoxin sorgoleone has been largely associated for allelopathic characteristics of sorghum. Due importance for sorgoleone production is not given in this study. It is suggested to study the QTL for sorgoleone content and yield. Further, QTL for sorgoleone needs to be associated with other components of allelopathic characteristics sorghum.

2. Validation of expression stability of QTL is missing in this study. It is suggested validate the identified QTL multiple seasons and location would be logical.

6. PLOS authors have the option to publish the peer review history of their article (what does this mean?). If published, this will include your full peer review and any attached files.

Reviewer #1: Yes: Mohamed El-Soda

Reviewer #2: Yes: Nagaraja Reddy

---

## [Author Response · Author response to Decision Letter 0]

2 Jun 2020

Review Comments to the Author

Response: Thank you very much for your time and giving your valuable comments and suggestions to uplift the standard of our manuscript. Our detail response to each comment of reviewers are given below.

Reviewer #1: The authors used two sorghum populations, i.e. SDRS and F2:3, to study the inhibitory effect of sorghum WSE on the seed germination and on the root length of lettuce seedlings. They used four concentrations of the WSE (25%, 50%, 75% and 100%). The authors identified several QTL in the F2:3 and in the association mapping populations. The work is designed in a proper way, however, I have the following comments;

Response: Thank you very much for your support and cooperation. We highly appreciate your valuable comments and suggestions for the improvement of our manuscript.

• I couldn't find any figures attached to the manuscript that could have helped me to better understand the experimental design.

Response: Sorry to know about that. The figure were attached but might have experienced some technical issue. I hope you will access in the revised version of manuscript. 

• Please, add how the heritability was calculated. Was the H2 for germ50% 0.07?

Response: Thank you for the comment and sorry for the typo, h2 for germ50% is 0.70.

Here, h2 was calculated as broad sense i.e. h2= Genetic Variance/Phenotypic variance 

Where Genetic variance = MS (Genotypes) – MS (Phenotype)/Total genotypes and 

Phenotypic variance = Genetic variance + Error Variance 

• Please, remove the last paragraph of the "Bioassay of allelopathic effect using water-soluble extract (WSE)" as it was explained again in the next paragraph "Statistical analysis". 

Response: Thanks for your suggestion, the paragraph has been removed in revised version of manuscript.

• For QTL mapping, was threshold 2.5 chosen based on 1000 permutation or as an arbitrary threshold?

Response: In QTL (Linkage) mapping, 2.5 threshold was chosen based on 1000 permutation using QTLcartographer. 

• I couldn't find any figure or data presenting the map of the F3 or the association panel. Please, provide this data.

Response: I really didn’t understand why but all figures were uploaded in previous version as well. I hope this time there will be no issue in viewing all these figures. 

Fig.2 is about Association mapping of 181 SSRs and 107 sorghum accessions.

Fig.3 is about Linkage mapping of 151 SSRs on 134 F2 population. 

Fig. 4 is showing Targeted association analysis of 22 QTL linked markes.

• In such low density map, I would recommend using window size larger than 10 cM, e.g. 30 cM, to consider the following sentence correct "The adjacent QTLs identified for the same trait with non-overlapping intervals on same chromosome were considered as different QTLs". In addition, one can consider two adjacent qtl as different ones only when their effect is different, e.g. one has an effect from A and the other from B.

Response: Thank you very much for you suggestion. It has been revised in new version of manuscript (Page # 8). No change reported in total identified QTLs reported in this study even if the window size of 30 cM is used. 

• Please change this title “Linkage disequilibrium (LD)-based association mapping” to “Population structure and association mapping” or add two subtitles “Population structure” and “Association mapping”

Response: Thanks, changed this heading in revised version to “Population structure and association mapping”.

• “the P-values representing the significance of LD was measured”, how was P value calculated? What is threshold of 2.5 and strict threshold =3 ? Please explain. Have you used an arbitrary threshold or Bonferroni correction? I believe a P value of 2.5 is relatively low and will increase the false positives.

Response: In Tassel software, MLM model identifies P-values based on nominal test of individual makers and then corrected for multiple testing. I used Benjaminin-Hochberg FDR method for corrected P-values. In this study we set the threshold of significance as –Log10 (P-corrected value) of 2.5 which is P<0.003 and more stringent threshold of –log10(corrected P-value) = 3.0 (i.e. P<0.001). Most of studies reported –log10(P)= 2.0 as threshold for quantitative traits which is equal to P value of 0.01. Even in some case a threshold of –log10(P)=1.3 (=pvalue 0.05) has been reported. As we believe allelochemical traits are complex and controlled by several QTLs with minor effects, therefore a threshold of –log10(corrected P-value) = 2.5 is quite reasonable. Also as mentioned earlier, the Benajimin-Hochberg correction is a strong tool for controlling fasle positives. 

https://tassel.bitbucket.io/docs/bradbury2007bioinformatics.pdf

• Last paragraph of Linkage disequilibrium (LD)-based association mapping, “As genome-wide association studies results in spurious associations in particular with low number of markers.” Not clear what Authors want to say.

Response: Actually as we know GWAS has major issue of false positives and especially if the number of markers used are low in number. To avoid this constraint, we used regional association mapping approach in which we just used the already identified QTL markers in this study and applied in association model. This further strengthened the reliability of our findings.

• In Marker localization and homology with known genes “The Map Viewer of NCBI website (http://www.ncbi.nlm.nih.gov/mapview/) was used to identify loci previously identified as linked to known genes in genome-based sequence information.” I don’t understand whether the author did that to know if their SSR were previously mapped, which could be done from previously published articles, or they did that to just know what is known. Please, clarify on that.

Response: Infact we wanted to localize our QTL markers with sorghum genome database and see if these loci have been previously linked to some known genes. We also tried to establish if our loci have any homology with other genes previously identified in sorghum for other related traits.

• In “Construction of linkage map……”, please, move the following sentence to discussion “These findings confirms the consistency and accuracy of the linkage map reported here.

Response: Thanks, the sentence has been removed.

• The discussion part, in general, needs to be restructured to avoid repeating sentences and contents. Below are some suggestions,

• when comparing the discrepancy in locating some SSR, e.g. SB3664 and SB2613, please give references after "in previous report".

Response: Thanks, done.

• The paragraph that starts with "We used two approaches and two different types of populations ……………..", it reads like a review not discussion. Can be deleted or modified.

Response: Thank you very much. The paragraph has been removed and did modification in next one. 

• "We used four different concentrations of WSE, ……………...." and then "This confirms...…....". It isn't clear what confirms what.

Response: Thanks, the sentence has been modified, “Agarwal et al. (2002), Iqbal et al. (2003), Fateh et al. (2012) and Shang & Xu (2012) also reported an increase in inhibitory effects with increasing concentration of allelochemicals.”

• It will be more coherent to gather all the discussion related to QTL and LD mapping in one paragraph, as now in several places I find the same conclusion, i.e. the suitability of the 2 approaches to map significant QTL ....."

Response: Thank you. We tried to make discussion more coherent and sequential in revised version. 

• Why discussing SOR1? Please take the space needed to elaborate on why specifically the authors discussed this gene

Response: Actually until recent, Sorgoleone is considered the only major allelochemical which controls more than 95% of allelochemical characteristics in sorghum and most of genetic research has been oriented on this chemical compound only. SOR1 is the only gene cloned and characterized that is controlling the release of sorgoleone. In our study we identified several other chromosomal regions that controls the release of many other allelochemicals in sorghum and are present in other parts as well rather than only in roots where sorgoleone is release. In revised version we emphasized more on the importance of SOR1 in context to our study.

• "A single common co-localized major QTL qtl10RL on Chr 10 between markers Xtxp270 (32.6 bp) and SB5329 (35.5 bp) was detected for inhibition of root length by WSE used in 50%, 75% and 100% concentrations in both approaches of QTL mapping". The content of this sentence was mentioned earlier in the discussion. Please, remove this sentence and all other repeated sentences.

Response: Thank you very much. Infact this QTL is identified as major one by two approached we used and also in targeted association. That’s why we emphasized more on it and this region could be important in isolation and cloning genes for allelochemicals in sorghum. However, we tried to polish the discussion and removed the repeated sentences from the text.

• The author should carefully interpret the results of the "regional mapping" and the physical co-location of with known genes as the QTL interval is normally large and contain several genes, especially with this low dense map. I would rather call them candidate genes

Response: Thanks and modified in revised manuscript.

Reviewer #2: Allelopathic characteristics of sorghum has been studied at large, however, the reports of QTL for allelopathic characteristics were not reported. In this line, the study “Genetic analysis of QTLs controlling allelopathic characteristics in sorghum” has a merit. Authors did very good job by conducting experiment in optimal way and results are also very interesting however authors need to focus on the results of the study. I propose to recommend for publication in PLOSONE with few suggestions to improve the manuscript.

Response: Thank you very much for your time and support. We highly appreciate your valuable comments and suggestions for the improvement of our manuscript.

Major revision

1. The phytotoxin sorgoleone has been largely associated for allelopathic characteristics of sorghum. Due importance for sorgoleone production is not given in this study. It is suggested to study the QTL for sorgoleone content and yield. Further, QTL for sorgoleone needs to be associated with other components of allelopathic characteristics sorghum.

Response: Thank you very much. We have added some more details about sorgoleone in introduction.

Sorgoleone is the only main phytotoxin that has been extensively studied in sorghum and is controlled by a single gene “SOR1” that has been completely characterized. There is no QTL study for allelopathy in sorghum until recent or atleast to our knowledge. This study gives more insight to the production of phytotoxins in sorghum other than sorgoleone. 

Actually, we have previously used same sets of SSR markers in an experiment on yield and yield components in Shehzad and Okuno (Euphytica 2015) but haven’t seen any major correlation between those QTLs and the ones reported in this study. We strongly agree with you about that QTL mapping for allelopathy should be associated with yield and hope to conduct such experiment in future.

2. Validation of expression stability of QTL is missing in this study. It is suggested validate the identified QTL multiple seasons and location would be logical.

Response: Sorghum diversity set used in this study was developed from genebank accessions and maintained in our lab since 2006. We have been growing these materials for many years in the field and maintaining its homogeneity. For Association study while using the SDRS, we selected seeds of same genotype harvested from different years and sown in pots. So there is less chance of skipping allelopathic affect due to environments. 

Another approach we adapted here to validate our result was using Targeted Association Mapping. As in this study the number of genotypes and markers are not too much high so to avoid false positives we tried regional association mapping. In this method we only used QTL loci identified in this study and performed association analysis. We have found four common QTLs that were located in GWAS and Linkage mapping approach. This shows the credibility of these results and can further be utilized in gene cloning experiments.

In case of linkage analysis, the F2 were sown in field conditions and the field of University of Tsukuba is properly maintained and organized. Also there are less variations in weather patterns in Tsukuba each year. After harvesting F3 seeds, 30 seeds from each F2 line were planted in pot conditions in controlled greenhouse so there is less chance of any environmental fluctuations. Although release of secondary metabolites are mainly controlled by genes but still environment affects its release and to avoid those environmental factors we have grown the material in green house and used them at seedling stage. Although for mapping complex traits more advance generations upto F7 or F8 are best but several studies reported the use of F3 populations for QTL mapping has similar power as of F7 or F8 generations (for example, Shohei Takuno et al, PLoS One: October 9, 2012). Also as mentioned earlier, use of multiple replications and repeats in bioassays also minimized environmental error.

Finally, we are planning to further proceed with these findings and fine map major QTLs identified in this study. This will ultimately help us cloning genes responsible for allelochemical characteristics in sorghum.

---

## [Decision Letter · Decision Letter 1]

11 Jun 2020

PONE-D-19-28312R1

Genetic analysis of QTLs controlling allelopathic characteristics in sorghum

PLOS ONE

Dear Dr. Shehzad,

Thank you for submitting your manuscript to PLOS ONE. After careful consideration, we feel that it has merit but does not fully meet PLOS ONE’s publication criteria as it currently stands. Therefore, we invite you to submit a revised version of the manuscript that addresses the points raised during the review process.

We look forward to receiving your revised manuscript.

Kind regards,

Craig Eliot Coleman, PhD

Academic Editor

PLOS ONE

Reviewers' comments:

Reviewer's Responses to Questions

**Comments to the Author**

1. If the authors have adequately addressed your comments raised in a previous round of review and you feel that this manuscript is now acceptable for publication, you may indicate that here to bypass the “Comments to the Author” section, enter your conflict of interest statement in the “Confidential to Editor” section, and submit your "Accept" recommendation.

Reviewer #1: (No Response)

2. Is the manuscript technically sound, and do the data support the conclusions?

Reviewer #1: Yes

3. Has the statistical analysis been performed appropriately and rigorously? 

Reviewer #1: Yes

4. Have the authors made all data underlying the findings in their manuscript fully available?

Reviewer #1: Yes

5. Is the manuscript presented in an intelligible fashion and written in standard English?

Reviewer #1: Yes

6. Review Comments to the Author

Reviewer #1: Authors have partially addressed the comments as they gave answers to the queries raised during the first round but only in their letter not in the main text. Please amend the manuscript based on the suggested changes. For example;

1. Please, add how heritability was calculated, and explain why you refer to SOR1 in the discussion part

2. Please, add to your methods how the threshold was chosen for QTL and association mapping. Please add R2 and how you calculated "FDR-corrected P value" for the association mapping, that might strengthen your point of using 2 as a threshold.

3. Please, check the language throughout the manuscript as many sentences are not correct. For example, "Each trait was analyzed with empirical experiment-wise threshold values for significance (P = 0.05) from estimating 1000 permutations"

4. In the QTL part, please discuss the QTL co-location as this will support your results.

5. Please rephrase the sentence you added to introduction, “Researchers studying secondary metabolites affecting germination of parasitic weed Striga asicatica (witchweed) first discovered it (Chang et al. 1986). Retrobiosynthetic NMR analysis was ................. ,

materials and methods, "We physically localized the SSR loci that were strongly linked with the traits by BLAST searches of sequences in http://www.phytozome.net/sorghum ……… ",

and to the discussion "also reported an increase in inhibitory effects with increasing concentration of allelochemicals. After statistical analyses, we detected 75% and 100% WSE as optimum --------", as they don’t read well.

7. PLOS authors have the option to publish the peer review history of their article (what does this mean?). If published, this will include your full peer review and any attached files.

Reviewer #1: Yes: Mohamed El-Soda

---

## [Author Response · Author response to Decision Letter 1]

13 Jun 2020

6. Review Comments to the Author

Thank you very much for the continuous support and help in revising our manuscript. 

Reviewer #1: Authors have partially addressed the comments as they gave answers to the queries raised during the first round but only in their letter not in the main text. Please amend the manuscript based on the suggested changes. For example;

1. Please, add how heritability was calculated, and explain why you refer to SOR1 in the discussion part

Response: Thanks, added in revised manuscript. The method of calculating heritability is added in methods under to heading Statistical Analysis and referring SOR1 in discussion part. Also Table 4. is modified by adding R2 values in new column.

2. Please, add to your methods how the threshold was chosen for QTL and association mapping. Please add R2 and how you calculated "FDR-corrected P value" for the association mapping, that might strengthen your point of using 2 as a threshold. 

Response: Thank you very much. We have added the method of FDR-corrected in material and methods and also R2 values for QTLs detected using GWAS and Targetted Association Analysis. The method of FDR corrected P value has been explained in material and methods under heading of Population Structure and association mapping.

3. Please, check the language throughout the manuscript as many sentences are not correct. For example, "Each trait was analyzed with empirical experiment-wise threshold values for significance (P = 0.05) from estimating 1000 permutations"

Response: Thank you very much. The manuscript has been gone through English editing once again by my colleague a native speaker and checked for linguistic errors.

4. In the QTL part, please discuss the QTL co-location as this will support your results.

Response: Thanks, we have already added the one found using several pipelines and added in the results and discussion part.

5. Please rephrase the sentence you added to introduction, “Researchers studying secondary metabolites affecting germination of parasitic weed Striga asicatica (witchweed) first discovered it (Chang et al. 1986). Retrobiosynthetic NMR analysis was ................. ,

materials and methods, "We physically localized the SSR loci that were strongly linked with the traits by BLAST searches of sequences in http://www.phytozome.net/sorghum ……… ",

and to the discussion "also reported an increase in inhibitory effects with increasing concentration of allelochemicals. After statistical analyses, we detected 75% and 100% WSE as optimum --------", as they don’t read well.

Response: Thanks, done.

---

## [Editor Report · Decision Letter 2]

25 Jun 2020

Genetic analysis of QTLs controlling allelopathic characteristics in sorghum

PONE-D-19-28312R2

Dear Dr. Shehzad,

We’re pleased to inform you that your manuscript has been judged scientifically suitable for publication and will be formally accepted for publication once it meets all outstanding technical requirements.

Kind regards,

Craig Eliot Coleman, PhD

Academic Editor

PLOS ONE
---

## [Editor Report · Acceptance letter]

30 Jun 2020

PONE-D-19-28312R2 

Genetic analysis of QTLs controlling allelopathic characteristics in sorghum 

Dear Dr. Shehzad:

I'm pleased to inform you that your manuscript has been deemed suitable for publication in PLOS ONE. Congratulations! Your manuscript is now with our production department. 

Kind regards, 

on behalf of

Dr. Craig Eliot Coleman 

Academic Editor

PLOS ONE